# Relapse of Neonatal *Escherichia coli* Meningitis: Did We Miss Something at First?

**DOI:** 10.3390/children8020126

**Published:** 2021-02-10

**Authors:** Nadja H. Vissing, Mette B. Mønster, Sannie Nordly, Gholamreza K. Dayani, Sofie S. Heedegaard, Jenny D. Knudsen, Ulrikka Nygaard

**Affiliations:** 1Department of Pediatrics and Adolescent Medicine, Copenhagen University Hospital, Rigshospitalet, 2100 Copenhagen, Denmark; mette.bondo.moenster@regionh.dk (M.B.M.); ulrikka.nygaard@regionh.dk (U.N.); 2Department of Pediatrics and Adolescence, Copenhagen University Hospital, 2650 Hvidovre, Denmark; sannie.brit.nordly@regionh.dk; 3Department of Pediatrics and Adolescence, Zealand University Hospital, 4000 Roskilde, Denmark; gday@regionsjaelland.dk; 4Department of Pediatrics and Adolescence, Herning Hospital, 7400 Herning, Denmark; sofie.sommer.hedegaard@vest.rm.dk; 5Department of Clinical Microbiology, Copenhagen University Hospital, Rigshospitalet, 2100 Copenhagen, Denmark; Inge.Jenny.Dahl.Knudsen@regionh.dk

**Keywords:** neonate, meningitis, *E. coli*, lumbar puncture

## Abstract

Relapse of neonatal meningitis is most commonly caused by *Escherichia coli*. Management to prevent relapse varies and evidence is limited. We present four cases of relapsing neonatal *E. coli* meningitis in Denmark in 2016–2017 and review the current literature on this subject. During the primary episodes, our patients received cephalosporin for 3 weeks and gentamicin for the first 3 days. The only identified risk factor was delayed CSF sterilization in three of four cases and no repeated lumbar puncture. Relapse occurred after 2–28 days; one case with ventriculitis and one with empyema. Relapses were treated for 6–14 weeks with monotherapy. No children had an underlying disease predisposing to *E. coli* meningitis. There is generally a trend towards reducing invasive procedures, e.g., lumbar puncture and the length of intravenous antibiotics in pediatric infectious diseases, but our cases highlight a condition where the opposite might be needed.

## 1. Introduction

Relapse of neonatal meningitis is most commonly caused by *Escherichia coli*, occurring in 2–21% of infants despite potent antibiotics [1,2], but risk factors are scarcely described. Management of neonatal *E. coli* meningitis to prevent relapse varies and the evidence is limited. We present four cases of neonatal *E. coli* meningitis with relapse and discuss strategies for prevention.

## 2. Cases

The four infants with relapse *E. coli* meningitis were diagnosed in Denmark in 2016–2017; a country with approx. 60,000 live births per year. The cases are summarized in Table 1 and detailed in Appendix A. The *E. coli* isolates were susceptible to the initial therapy, including third generation cephalosporins and gentamicin. During the primary episodes, the first lumbar punctures were performed 20–58 h after initiation of targeted therapy, and three were still culture positive despite ongoing treatment. None of the three culture-positive cases had repeated CSF examination to secure sterilization. Cerebral magnetic resonance imaging (MRI) during or shortly after cessation of therapy was without signs of persistent infection, and abdominal ultrasound with no urinary tract malformations. All cases were treated for 3 weeks in total. The relapse *E. coli* episodes occurred 2–28 days after completion of the initial 3 weeks therapy, all with isolates with an identical susceptibility pattern, and case 3 with identical strain shown by whole genome sequencing. There was no evidence of ESBL (extended spectrum beta-lactamase) production. At relapse, MRI was abnormal in two cases showing signs of ventriculitis (case 3) and empyema (case 4), while case 1 and 2 had normal MRI. The relapses were treated for 6–14 weeks. No children had an underlying disease predisposing to *E. coli* meningitis. 

At follow-up, all children were healthy, although case 3 had moderately delayed developmental milestones without other signs of neurological disease.

## 3. Discussion

We present four cases of relapsing neonatal *E. coli* meningitis despite 3 weeks of targeted treatment. There were no apparent risk factors during the primary meningitis episode, except for persistent bacterial growth in the CSF 1–2½ days after initiation of antibiotics in three of four cases and lack of repeated lumbar puncture to secure sterilization.

The need to perform a repeat lumbar puncture is a frequent consideration for the neonatologist, and recommendations vary due to limited evidence. Standard textbooks recommend a repeated lumbar puncture 48–72 h into treatment of *E. coli* meningitis to document CSF sterilization and provide reassurance of effective therapy in infants with this particularly virulent organism and guide treatment duration [3,4]. The UK guidelines do not recommend repeat lumbar puncture in neonates with good clinical recovery [5]. In fact, many clinicians do not routinely perform this invasive procedure in a stabilized infant [6,7]. In this study, all four children were treated with adequate antibiotics and showed good clinical recovery, but relapse still occurred. None of the three culture-positive cases had repeated lumbar puncture performed, and their relapses may have been caused by delayed sterilization of CSF and an unfulfilled need for longer duration treatment. 

Our findings encourage the use of repeat lumbar puncture, although we acknowledge that (1) other factors, not identified, could have been involved in the relapses, and (2) the association between persistent CSF growth and subsequent relapse has not been addressed systematically. However, studies have shown an increased risk of complications and death in those with positive repeat lumbar puncture, found in 10–15% of cases [7,8,9,10]. Delayed CSF sterilization should lead to prolonged antibiotic therapy [1,2], but may also warrant consideration of increasing the dose to achieve higher CNS concentrations, changing the antibiotic to a drug with lower minimal inhibitory concentration (MIC), or even adding an additional antibiotic. Furthermore, cerebral imaging should be considered, since the persistence of bacterial growth can indicate a purulent focus (e.g., ventriculitis) that may require surgical intervention or increased duration of antimicrobial therapy [3].

Combination antibiotic therapy in primary *E. coli* meningitis has been suggested for improving treatment, e.g., adding ciprofloxacin, which has good penetration to CSF and cerebral tissue, as well as bactericidal activity and low MIC against *E. coli*. However, no randomized trials have explored this, but a retrospective study showed that first-line adjunct ciprofloxacin to neonates with primary *E. coli* meningitis did not decrease the proportion of infants with CSF sterilization failure after a median of 49 h of initiation of therapy (11% vs. 14%), and ciprofloxacin did not improve neurological outcome or mortality [10]. Gentamicin was added as initial empiric therapy in all our cases, as often recommended in septic neonates, but this did not prevent relapse. 

Lumbar puncture at the end of therapy is generally not recommended in uncomplicated cases since abnormal CSF findings often persist irrespective of successful treatment [3]. 

Neuroimaging is also used to identify children at risk of complications, which may require prolonged treatment or surgery, i.e., ventriculitis, cerebral abscess, empyema, or persistent parameningeal foci. It is recommended shortly before cessation of antibiotics, even in uncomplicated courses. In all our cases, cerebral MRI during or shortly after the primary episode was without signs of persistent infection and not indicative of relapse. This suggests that the patients harbored residual infectious foci within the brain substances or paraventricular regions not evident on MRI, illustrating that a normal MRI does not exclude subsequent relapse. However, three of the MRI’s were performed without contrast, which may have limited visualization of persistent infection.

All children had initial high levels of C-reactive protein (mean 201 mg/L, range 165–260) with fast decline within a few days after treatment initiation. All children had normal C-reactive protein at termination of antibiotic treatment. Procalcitonin was not measured systematically. We therefore do not have data to support that inflammatory markers followed an unusual pattern or that normal patterns can exclude relapse. Hopefully, future research will identify biomarkers that can monitor persistent silent infection. 

Some underlying conditions predispose to neonatal *E. coli* infection. Urinary tract malformation is a risk factor for *E. coli* meningitis [2] and thus a potential reservoir for relapse, and ultrasound of the urinary tract is indicated in primary neonatal *E. coli* meningitis or sepsis. All our cases had normal urinary tract anatomy. 

Primary immunodeficiency has not been associated with neonatal *E. coli* meningitis, relapse or recurrence. The susceptibility to *E. coli* infection in neonates is well explained by their immature immune system including a poor antibody response to the K1 capsular antigen carried by the majority of *E. coli* strains. Transitional neutropenia is common in Gram-negative bacterial meningitis, but neonatal *E. coli* meningitis has not been reported to occur particularly in children with congenital neutropenia or other phagocyte disorders. Thus, routine testing for primary immunodeficiency is not indicated in case of neonatal *E. coli* meningitis or relapse.

Dermal sinus tracts malformation is a well-described risk factor to *E. coli* meningitis, and a careful physical examination of the spine is essential, particularly in episodes which re-occur after a longer interval (e.g., 3 weeks). None of our cases had spinal malformations.

Galactosemia is rare hereditary disorder of carbohydrate metabolism predisposing to neonatal *E. coli* infection and should therefore be considered in the case of invasive *E. coli* infection [11]. Screening for galactosemia is included in some newborn screening programs or can be specifically tested. Also, children with galactosemia are unable to tolerate mother’s milk and will normally present early in life with failure to thrive and elevated liver enzymes. None of our children had galactosemia. 

None of our cases were of full-term gestation. Three were premature, and one was born ‘early term’ (GA 37 + 1). This is consistent with the existing literature showing *E. coli* meningitis to be 7-fold more frequent in preterm neonates [9]. To our knowledge, it has not been explored if prematurity is related to an increased risk of relapse per se, but it is likely that children who are not born at full-term gestation are also at higher risk of relapse of their meningitis. Thus, a higher vigilance could be considered for these patients.

All cases presented were interpreted as relapse of the primary infection in contrary to recurrent meningitis, which is traditionally defined as a secondary meningitis occurring more than 21 days after cessation of therapy, often with a different organism [12]. In such cases, cranial anatomical defects and primary immunodeficiency should be ruled out [13]. The secondary episode in case 3 did in fact occur 28 days after cessation of therapy, but the *E. coli* strain was shown to be identical by whole genome sequencing.

### Treatment

Treatment of relapsing *E. coli* meningitis in children is very sparsely reported, and guidance for the clinician is minimal, including choice of antibiotics and duration of treatment. Our cases, including one with ventriculitis and one with empyema, received prolonged treatment (8–14 weeks) decided by the treating physician, without additional relapse. In case 4, it was chosen to treat with additional ciprofloxacin for the first 3 weeks.

Our cases demonstrate that neonatal *E. coli* meningitis is a complex disease with risk of relapse. If possible, early involvement of a specialist in Pediatric Infectious Diseases is advisable. Figure 1 summarizes important clinical key points to consider when treating an infant with *E. coli* meningitis.

## 4. Conclusions

These cases illustrate the risk of relapsing infection in neonates with *E. coli* meningitis. There were no clinical clues except delayed CSF sterilization in three of four cases. Our findings support the use of repeat lumbar puncture even in stable children to ensure CSF sterilization and guide treatment. As there is generally a trend towards reducing invasive procedures and the length of intravenous antibiotics in pediatric infectious diseases, our cases highlight a situation where the opposite might be needed.

## Figures and Tables

**Figure 1 children-08-00126-f001:**
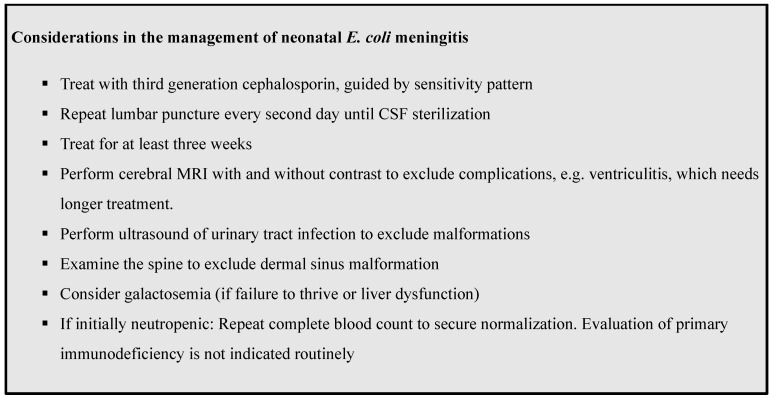
Key Clinical Points.

**Table 1 children-08-00126-t001:** Cases of relapse of neonatal *Echerichia coli* meningitis.

Case				First *E. coli* Meningitis				Relapse *E. coli* Meningitis		Follow-Up		
	Sex	GA at birth (weeks)	BW (gram)	Age (days)	Treatment	Cerebral MRI	Urinary ultra-sound	Time from completion (days)	Treatment	Age (years)	Under-lying disease	Sequelae *
1	F	36 + 3	2130	3	Cefotaxime 21 days + gentamicin day 1–5	Minor hemorrhagic parenchymal infarctions. No signs of infection. (without contrast)	Normal	4	Meropenem 1 week followed by ceftriaxone 9 weeks	4.0	No	No
2	M	34 + 0	1980	5	Cefotaxime 21 days + gentamicin day 1–3	Normal (with contrast)	Normal	11	Cefotaxime 1 week, meropenem 1 week followed by ceftriaxone 12 weeks	3.5	No	No
3	M	34 + 5	2280	45	Cefotaxime 20 days + gentamicin day 1–7	Ischemic changes of cortical parenchyma. No signs of infection. (without contrast)	Normal	28	Ceftriaxone 8 weeks	3.5	No	Moderate sensory disabilities, visual impairment, delayed development. No signs of hydrocephalus or cerebral palsy. Persistent changes on MRI.
4	F	37 + 1	2940	13	Cefotaxime 22 days + gentamicin day 1–3	Minor hemorrhagic parenchymal infarctions. No signs of infection. (without contrast)	Normal	2	Meropenem + ciprofloxacin 6 weeks followed by ceftriaxone 3 weeks	4.0	No	No

F = Female; M = Male; GA = Gestational Age; BW = Birth Weight; MRI = Magnetic Resonance Imaging. * Developmental milestones and audiology were investigated, in addition to signs of cerebral palsy and hydrocephalus.

## Data Availability

Further details on the patients can be accessed by contacting the corresponding author.

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
