# Peer review of "Relapse of Neonatal Escherichia coli Meningitis: Did We Miss Something at First?"

_children, 2021, doi:10.3390/children8020126_

Round 1

Reviewer 1 Report

This is in interesting collection of cases of recurrent E. coli meningitis in neonates. The cases and their discussion illustrate the need for better guidelines, some of which are suggested in this report. A few questions come to mind:

  1. Where laboratory markers used to guide length of treatment that could have indicated persistent infection/inflammation such as non-decreasing or increasing levels of C-reactive protein? Should trending of inflammation markers be added to the recommendations for management of E. coli meningitis?
  2. Were the brain MRIs performed with contrast? Including contrast is important to visualize complications from meningitis such as formation of an abscess or empyema. MRI with and without contrast should be added to the recommendations for management of E. coli meningitis.
  3. Was galactosemia ruled-out? Galactosemia is mentioned as a possible underlying risk factor but it is not mentioned if this was specifically ruled out in either case (e.g. normal newborn screening for inborn error of metabolism.
  4. Was there evidence for production of extended-spectrum β-lactamases (ESBLs). It has been described that use of third-generation cephalosporins can induce the expansion of ESBL-producing E. coli (e.g. Nakayama et al., FEMS Microbiology Letters 2018).
  5. Was Infectious Diseases consulted? These cases demonstrate the complexity of this disease and the importance for long-term follow up, which warrants early involvement of a specialist in Pediatric Infectious Diseases.

Reviewer 2 Report

Brief summary:

This is a nice article on relapsing E.coli meningitis in the neonatal population in Denmark. The authors describe 4 cases and review the literature as well as give suggestions / considerations in the management of neonatal E. coli meningitis. A few comments I would like to make and would like the authors to address:

Comments:

  1. Regarding Case 1. Relapse occurred 4 days after completion of antibiotic therapy. CSF was culture negative yet with elevated WBC count. Given lack of repeat LP at the end of initial therapy and short time between end of therapy and relapse, how do you know the patient had meningitis this time? i.e. How do you know the elevated WBC in CSF was not from the prior infection as a repeat LP was not done at the end of therapy? How do you know this was a meningitis as nothing grew in CSF and no PCR was done? Is it possible that in case 1 the relapse was only in blood and there was no meningitis? I do not know the answer, I am just looking at the data and wondering if this can be considered a relapsing meningitis vs just sepsis the 2nd time?
  2. Regarding Case 4. Birth weight is listed as 2490 grams in the first table embedded in the text. But listed as 2940 grams in Table 1 (Appendix A). I would like authors to correct this wherever it is wrong.
  3. Regarding risk factors. Three of the infants were late preterm infants. Case 1 was born at 36+3 weeks of gestation. Case 2 was born at 34+0 weeks of gestation. Case 3 was born at 34+5 weeks of gestation Even case 4 even though term was born at 37+1 weeks of gestation (early term). None of the 4 patients with relapsing E. Coli meningitis were of full term gestation. Authors should discuss this and perhaps add “absence of full term gestation” (or similar description) to the list of “Considerations in the management…” as one of the risk factors to consider. i.e. in an infant with E. Coli meningitis who was not born at full term gestation a higher vigilance should be considered for duration of therapy, repeat LP, etc.
    1. Similarly, if birth weight of case 4 (see comment #2 above) is truly 2490 grams this would make that infant small for gestational age (SGA, <10% ile). Being late preterm and/or SGA could be a risk factor for meningitis. If birthweight is 2940 grams then this would not be the case.
  4. Regarding repeat LP. Authors suggest a repeat LP every second day until CSF sterilization. That seems to be quite often as it is an invasive procedure. If one plans to treat 21 days, and initial CSF culture shows E. Coli sensitive to a specific antibiotic, and antibiotic given matches it, would a repeat culture slightly later (e.g. 72h or 96h into antbiotic course) be acceptable to avoid too many repeat LPs?
  5. Regarding Figure 1 “Key Points”. I really like this table authors compiled regarding management considerations, a nice summary.

 If above comments/suggestions are considered (broad and specific comments), I think this study will add to the field and be welcomed by the readers of Children.
